# Determinants of contraceptive use among women 0–23 months postpartum in Kitui County, Kenya: A cross-sectional study

**Lilian Mutea**[1]*, **Immaculate Kathure**[1], **Damazo T. Kadengye**[2], **Sila Kimanzi**[1], **Daniel Wacira**[1], **Nelson Onyango**[3], **Hesborn Wao**[2]

**1** US Agency for International Development (USAID), Nairobi, Kenya, East Africa, **2** Africa Population and Health Research Centre (APHRC), Nairobi, Kenya, **3** School of Mathematics, University of Nairobi, Nairobi, Kenya

* Lmutea@usaid.gov

## Abstract

The risk of unintended pregnancy is high in the postpartum period, especially during the first year of delivery. Yet, short birth intervals are associated with increased risk of adverse maternal and infant outcomes. In Kenya, despite women having multiple contacts with healthcare providers during their pregnancy and postpartum journeys, uptake of contraceptives during the postpartum period remains low. We examine factors that determine contraceptive use among postpartum women in Kitui County, Kenya.A cross-sectional study was conducted in six sub-counties of Kitui County covering a random sample of 768 postpartum women in April 2019. Logistic regression was used to study the association between uptake of contraceptives among women 0–23 months postpartum and several explanatory variables that included socio-demographic characteristics and facility-level factors. Overall, 68% of women in Kitui County reported using contraceptives. The likelihood of contraceptive use increased with the increase in the number of known family planning methods. Women who discussed family planning with a health worker within the last 12 months were 2.58 (95%CI: 1.73, 3.89) times more likely to use contraceptives during the postpartum period compared to those who did not. There was an increased odds of contraceptive uptake among women who received family planning information or service during postnatal care than those who did not (aOR = 2.04, 95%CI: 1.30, 3.24). A positive association was also found between contraceptive use and receipt of family planning information or service during immunization visits or during child well visits. It is evident that facility-level factors such as discussing family planning with women; educating women about different family planning methods; providing family planning information or services during postnatal care, immunization, or well child visits are associated with increased likelihood of contraceptive uptake by women during postpartum period. Programs targeting enhancing women's attendance of postnatal care clinics should be encouraged.

**Data Availability Statement:** All relevant data used in this paper have been included within the paper. Additional approvals will be required from the

ethical review committees for applications to re-use the transcript data, and in keeping in line with informed consent and assent agreements. Applications can be submitted to the Kenya Medical Research Institute research ethics reviews committee at seru@kemri.org.

**Funding:** Data used in this study were collected as part of a USAID Kenya and East Africa funded program AID-615-A-17-00004, awarded to Jhpiego. The role of The funding agency was restricted to providing oversight to program implementation and efficient use of resources. While some of the authors are employees of USAID which is the funding agency, the authors did not have a role in influencing the study design, development of tools, data collection or analysis. The time spent in development of the manuscript was paid by USAID and other author affiliate organizations (APHRC and University of Nairobi), and not by the awardee, Jhpiego. The funders had no role in study design, data collection and analysis, decision to publish, or preparation of the manuscript.

**Competing interests:** The authors have declared that no competing interests exist.

## Introduction

The risk of unintended pregnancy is high in the postpartum period, especially during the first year of delivery [1, 2]. For instance, short birth intervals (<15 months) are associated with increased risk of adverse maternal and infant outcomes including induced abortions, low birth weights, miscarriage, preterm births, neonatal or infant mortalities, stillbirths, and maternal depletion syndrome [3, 4]. The World Health Organization (WHO) recommends birth intervals of between 2 and 3 years [5]. Spacing births by at least 2 years can reduce maternal and child mortality by over 30% and 10%, respectively [4, 6], and three years' spacing reduces under-five mortality by 25% [7]. Women can space births by initiating the use of modern contraceptive methods within the first 12 months following delivery, also referred to as postpartum family planning [8, 9].

Whereas the majority (91%) of postpartum women in low- and middle-income countries (LMIC) desire to have a space birth by at least a year [10], contraceptive uptake continues to be low in this region [11, 12]. A recent systematic review and meta-analysis characterizing contraceptive use among postpartum women in LMICs established low postpartum family planning (FP) use and high unmet need for contraception following pregnancy [13]. Recent studies have established some of the reasons women desire to space births by at least a year. A qualitative study carried out in Malawi among post-partum women who had experienced a poor obstetric outcome indicated that women who lost a child after delivery did not wait long before conceiving, while those with normal deliveries desired to delay conception so that their bodies could recuperate, others delayed conception due their bad experiences in their last pregnancy i.e. having had cesarean or high blood pressure (1), yet another study in Uganda established that women had a clear understanding of the benefits of adequate child spacing, like allowing their bodies to recover and children getting proper nutrition and time to grow before another pregnancy, but this knowledge was difficult to translate into practice as women are disempowered to exercise child spacing. Women who used child spacing without their husband's consent risked losing financial and social assets (2). Whereas the need for contraceptives varies during a woman's reproductive years, the need is elevated during the postpartum period [14]. In Kenya, despite women having multiple contacts with healthcare providers in health facilities (e.g., during antenatal care [ANC], postnatal care [PNC], or immunization visits), uptake of contraceptives among postpartum women remains low. This may be due to limited choice of contraceptives, inability to access appropriate contraceptives, or lack of knowledge about the resumption of menses and return of fertility [15]. Other determinants of contraceptive use among postpartum women include level of education, nature of partner communication, and perceived need of family planning [16]; social and cultural factors [17]; and lower age, being married, higher education level, being employed and getting contraceptives at a health facility [18].

In Kenya, government health facilities present an opportunity for postpartum women to receive contraceptives, especially long-acting reversible contraceptives [18]. Kenya's health system is devolved whereby the national government focuses on policy development whereas the 47 county governments independently decide how to translate policies into practice including health service delivery. Provision of family planning (FP) services is guided by the national government via the national FP guidelines (6<sup>th</sup> edition) for service providers and FP standards of care. At the service delivery points in health facilities, women of reproductive age (15–49 years) receive contraceptive information, training on the use of WHO decision making tools for FP clients and providers and defining problems, needs and information gaps. Consequently, determinants of health service utilization, including postpartum contraceptive use, may vary across counties. While the overall contraceptive prevalence rate (CPR) in Kitui

County is 55%, the postpartum contraceptive use is not well known. Establishing the contraceptive use levels, patterns and factors that affect postpartum contraceptive use in specific settings is critical to informing strategies to enable women to effectively space and limit pregnancies and improve their overall health and that of their child. The purpose of this study was to examine the factors that determine contraceptive use among postpartum women (0–23 months) in Kitui County, Kenya. The study findings are expected to inform design of interventions aimed at increasing contraceptive use among postpartum women 0–23 months.

## Data and methods

### Ethics statement

The study protocol was approved by two Institutional Review Boards (IRBs)–KEMRI (IRB No. 597) and John Hopkins School of Public Health (IRB No. 00008364). Informed consent was obtained in writing before data collection began. Even though adolescents aged 15–19 years were part the study population (15–49 years), they are classified by the Ministry of health as mature minors by virtue of having been pregnant and/or given birth and therefore able to give informed consent.

### Permission for use of data

The study protocol was approved by two Institutional Review Boards (IRBs)–KEMRI (IRB No. 597) and John Hopkins School of Public Health (IRB No. 00008364). The research team was provided with the ethics approval letter, and they adhered to the study protocol. This was done through ensuring voluntary participation, and that informed consent was obtained in writing before data collection began. The study team also ensured that privacy and confidentiality measures were upheld during the data collection process. Maintaining data confidentiality of data collection materials and information after data collection, limiting access of study information to only the authorized personnel and excluding personal identifying information during data analysis and reporting.

### Study setting

A cross-sectional study design was employed whereby quantitative data were collected using a structured household survey questionnaire administered in April 2019. The study was conducted in Kitui County, one of the 47 counties located in the eastern part of Kenya. Kitui County has about 1.13 million people, with a population density of 37 people per km$^2$ [19]. It is located in arid and semi-arid land and is characterized by relatively high levels of poverty. The average distance to a health facility is 10.2km and 5% of the population live within 1 km of a health facility, 20% between 1-5km and 75% 5km or more [20]. There are 172,300 children under five years and 246,300 Women of Reproductive Age (WRA), constituting 16% and 23% of the total population respectively. There are 322 health facilities in Kitui, 15 hospitals (11 public and 4 private), 56 health centers, 241 dispensaries and clinics and 10 nursing homes. Two main public referral hospitals are the Kitui County Referral Hospital and the Mwingi General Hospital.

### Study procedures

A survey questionnaire was administered to mothers enrolled in the study. Data were collected on maternal socio-demographic characteristics; pregnancy months at first ANC; number of ANC visits; whether FP information or service was received during delivery, ANC, PNC, or immunization visits; place of delivery; assistance with delivery; number of family planning

methods known; whether FP was discussed in the last 12 months; and pregnancy intent. Mothers self-reported their contraceptive use. In this study, postpartum contraceptive use is defined as the uptake of modern contraceptives within a year of delivery. Quantitative data were collected using android enabled phones and electronically captured through the REDCap software. Collected data were automatically sent to a central server. Electronic data collection was preferred as it limits errors in data-entry.

The data used in this study were part of a larger study aimed at establishing baseline level for the utilization of high quality, county led family planning, reproductive, maternal newborn, child and adolescent health [20]. This assessment covered six of the eight sub-counties of Kitui County (i.e., Kitui Central, Kitui East, Kitui South, Mwingi Central, Mwingi North and Mwingi West). The county and sub-counties were the project implementation sites. The target group for this study were women 0–23 months postpartum. Lot quality assurance sampling (LQAS), a rapid, low-cost and practical random sampling methodology, was used to determine the sampling frame [21]. A total of 768 women from 32 wards in targeted sub-counties were identified. Each of the wards was allocated 24 data points for household interviews. In order to select the number of villages per ward and the number of household per village, a list of villages was drawn from each of the wards and probability proportional to size (PPS) sampling was used to allocate the data points to the villages, while a skip pattern was applied to identify households for interview in the villages, however if the selected the household did not have an eligible mother-child (under five) pair the interviewer moved to the next household for the interview. This approach ensured equal representation of households selected for interview among the villages. To ensure that the relevant information on each key indicator was collected and allow coverage to be reported at sub-county levels, the assessment was split into two modules: 1) demographics module and 2) maternal, child health and family planning module that targeted women who are mothers of children 0–59 months of age.

The structured household survey questionnaire used to collect data was developed by the health team using a reference questionnaire protocol from Kenya Medical Research Institute (KEMRI). The questionnaire was developed in English and then translated into Kamba, the predominant language spoken in the study setting. The questionnaire was pilot tested in some non-sampled wards in Kitui Central. Research assistants were trained on how to administer the questionnaires.

## Variables

The dependent variable was contraceptive use among women 0–23 months postpartum (scored 1 if mother reported using a specified modern method of contraception, else scored 0). Women who reported using traditional methods such as abstinence, withdrawal and lactational amenorrhea were classified as non-users of contraceptive methods. Independent variables included sub-county or ward (6 categories); education level (5 categories); religion (5 categories); marital status (2 categories); mother's source of income (7 categories); spouse level of education (6 categories); spouse's occupation (8 categories); income decision-maker (5 categories); health decision-maker (5 categories); mother's age (continuous); number of FP methods known (discrete); specific FP methods known; specific FP method used; knowledge of where to get FP service; whether a mother discussed FP with a HCW in last 12 months; whether a mother received FP information/messages during ANC; whether a mother received specified FP service at delivery; whether a mother received specified FP service at pre-discharge; whether a mother received specified FP at PNC; whether a mother received specified FP service at immunization; whether a mother used FP after immunization services; whether a mother received FP at Well-Baby Clinic; and whether the mother received FP services after visiting well child Clinic.

## Data management and analysis

Data were retrieved from the REDCap and exported as an Excel CSV file into R for statistical software [22] for analysis. Descriptive statistics including frequency and percentages (for categorical variables) and means, and standard deviation (for continuous variables) were computed. Logistic regression was used to study the relationship between the dependent variable (uptake of postpartum contraceptive) and the explanatory variables, first at bivariate level, and then at multivariable level in order to control for other variables. Statistical significance was set at 5%.

## Results

### Participant characteristics

Table 1 shows the distribution of women's characteristics by contraceptive use during postpartum. Overall, 427 (68%) women reported using contraceptives during postpartum, with Mwingi North and Kitui South having slightly lower than the average proportions of contraceptive users at 61% and 65% respectively. Younger women (aged 15–24 years) and older women (aged 35–49 years) had lower proportions among contraceptive users at 64% and 62% respectively compared to youthful women (aged 25–34 years) at 72%. All women had attended antenatal care (ANC), however, those who went for their ANC after the first trimester or those who didn't know had lower proportions of postpartum contraceptive users respectively at 66% and 68%, compared to those who had their first ANC earlier (72%). More than 75% of women delivered at a public health facility (50.1% at public and 25.8% at private) and were attended to by a skilled healthcare provider. Further, women who attended less than three ANC visits, those who did not deliver at a health facility or those who were not attended to by a skilled provider, had lower proportions of postpartum contraceptive users. More than one-half of the women (56.4%) had never discussed family planning (FP) with a health worker in the last 12 months–and indeed had the lowest proportion of contraceptive users (59%) compared to those that had discussed FP (79%). Not surprisingly, women who knew fewer (0–3) FP methods have the lowest proportion of contraceptive users (44%). However, there appeared to be no trend among women who knew four or more FP methods. Less than half of all the women reported to have received FP information or service during almost all service visits points namely, ANC visits (53.6%), delivery visit (44.5%), PNC visits (47.4%), immunization visits (46.1%), and during well child visits (42.9%). Further, women who received FP information or service during the different service point visits seem to have higher than the average proportion of postpartum contraceptive use, compared to those who did not receive FP service.

### Factors associated with contraceptive use

We found no statistically significant relationship between postpartum uptake of contraceptive and age of mother or pregnancy months at first ANC visit. Table 2 shows that women who had less than four ANC visits were less likely to use contraceptives (OR = 0.60, CI: 0.42, 0.86), however, this statistically significant relationship disappeared when other factors were controlled (aOR = 0.72, CI: 0.48, 1.08). No statistically significant relationship was found between contraceptive uptake and receipt of FP information during delivery, place of delivery, and assistance with delivery. The likelihood of contraceptive use increased with the increase in the number of FP methods known. For example, women who knew 7–9 FP methods were 2.78 times more likely to use contraceptives than those who knew 0 to 3 FP methods. Those who knew 10–12 FP methods were 3.12 times more likely to use contraceptives than those who knew 0–3 FP methods. Women who discussed FP with a health worker within the last 12

**Table 1. Frequency distributions of women characteristics by postpartum contraceptive use.**

| Variable | Classification | Non-user (n = 204) | User (n = 427) | Total (N = 631) |
|---|---|---|---|---|
| Sub-County | | | | |
| | Kitui Central | 30 (29%) | 74 (71%) | 104 |
| | Kitui East | 31 (29%) | 76 (71%) | 107 |
| | Kitui South | 46 (39%) | 72 (61%) | 118 |
| | Mwingi Central | 38 (32%) | 82 (68%) | 120 |
| | Mwingi North | 35 (35%) | 64 (65%) | 99 |
| | Mwingi West | 24 (29%) | 59 (71%) | 83 |
| Age of mother (in years) | | | | |
| | 15–24 | 68 (36%) | 123 (64%) | 191 |
| | 25–34 | 94 (28%) | 236 (72%) | 330 |
| | 35–49 | 42 (38%) | 68 (62%) | 110 |
| Pregnancy months at first ANC | | | | |
| | 1–3 months | 41 (28%) | 106 (72%) | 147 |
| | 4–9 months | 157 (34%) | 308 (66%) | 465 |
| | Don't Know | 6 (32%) | 13 (68%) | 19 |
| Number of ANC visits | | | | |
| | ≤3 visits | 82 (40%) | 124 (60%) | 206 |
| | 4+ visits | 114 (28%) | 287 (72%) | 401 |
| | Don't Know | 8 (33%) | 16 (67%) | 24 |
| Place of delivery | | | | |
| | Home/Other | 57 (38%) | 95 (63%) | 152 |
| | Private | 48 (29%) | 115 (71%) | 163 |
| | Public | 99 (31%) | 217 (69%) | 316 |
| Assistance with delivery | | | | |
| | Skilled | 144 (30%) | 331 (70%) | 475 |
| | Unskilled | 60 (38%) | 96 (62%) | 156 |
| Number of FP methods known | | | | |
| | 0–3 | 15 (56%) | 12 (44%) | 27 |
| | 4–6 | 56 (26%) | 158 (74%) | 214 |
| | 7–9 | 45 (44%) | 58 (56%) | 103 |
| | 10–12 | 88 (31%) | 199 (69%) | 287 |
| Discussed family planning in the last 12 months | | | | |
| | No | 147 (41%) | 209 (59%) | 356 |
| | Yes | 57 (21%) | 218 (79%) | 275 |
| Received family planning information during ANC | | | | |
| | No | 89 (30%) | 204 (70%) | 293 |
| | Yes | 115 (34%) | 223 (66%) | 338 |
| Family planning service during delivery | | | | |
| | No | 124 (35%) | 226 (65%) | 350 |
| | Yes | 80 (28%) | 201 (72%) | 281 |
| Family planning service during PNC | | | | |
| | No | 133 (40%) | 199 (60%) | 332 |
| | Yes | 71 (40%) | 228 (76%) | 299 |
| Family planning service during immunization visit | | | | |
| | No | 125 (37%) | 215 (63%) | 340 |
| | Yes | 79 (27%) | 212 (73%) | 291 |
| Family planning service during well child visits | | | | |

(*Continued*)

**Table 1.** (Continued)

| Variable | Classification | Non-user (n = 204) | User (n = 427) | Total (N = 631) |
|---|---|---|---|---|
| | No | 130 (36%) | 230 (64%) | 360 |
| | Yes | 74 (27%) | 197 (73%) | 271 |
| Pregnancy intent | | | | |
| | No | 86 (33%) | 178 (67%) | 264 |
| | Undecided | 54 (39%) | 84 (61%) | 138 |
| | Yes | 64 (28%) | 165 (72%) | 229 |

Note: **n** = frequency; **%** = percent; **User** = Used contraceptive; **non-user** = did not use contraceptive; **FP** = Family Planning; **ANC** = Antenatal Care; **PNC** = Postnatal Care

months were 2.58 (95% CI: 1.73, 3.89) times more likely to use contraceptives compared to those who did not discuss FP. Whereas no statistically significant relationship was found between contraceptive uptake and receipt of FP information at ANC, there was an increased odds of contraceptive uptake among women who received FP information or service during PNC than those who did not receive FP information or service (OR = 2.04, 95%CI: 1.30, 3.24). A positive association was found between contraceptive use and receipt of FP information or service during immunization visits or during child well visits, however, the statistically significant relationship disappeared when other factors were considered, as shown on Table 2 below.

## Discussions and conclusion

This study sought to establish determinants of contraceptive use among women 0–23 months postpartum in Kitui County, Kenya. Several potential factors perceived to influence contraceptive use in the postpartum women were examined. These factors, broadly categorized into demographic, reproductive characteristics, facility-based health services, and contraception availability and acceptability [13], are discussed next.

The finding that age of mother is not statistically significantly associated with postpartum uptake of contraceptive contradict findings from a prior study in which a significant relationship was found between age and contraceptive use [23]. We suspect this contradiction may be due to the relatively small number of women in higher age groups compared to women in lower age groups. Alternatively, it may be that the women in higher age groups, being more socially conservative, declined to be surveyed. Similarly, we found no association between pregnancy intent, our proxy for fertility intention (e.g., desire to space or limit pregnancy) and postpartum contraceptive use. However, in several studies, contraceptive use has been found to be significantly higher among women who report wanting to limit or space their next births compared to women who report having a desire to bear more children [10, 24–26].

The finding that mothers who have less than four attendances at ANC clinics are less likely to use contraceptives compared to those who have four or more visits (OR = 0.60, 95%CI: 0.42, 0.86) is consistent with findings from prior studies in Kenya [27, 28]. Noteworthy, ANC clinics are part of the critical service points in a health facility where mothers access information on FP. Whereas attendance at ANC clinics is recommended for initiation during the first trimester of pregnancy by WHO, part of our study findings indicate that contraceptive use is higher among women who initiate their clinic attendance four months or later. This means that there is more exposure to FP messaging by health care workers as evidenced by the finding that there were elevated odds of contraceptive use among women who discussed FP options in the last 12 months with HCWs than those who did not. Early ANC attendance ensures positive pregnancy outcomes by providing ample time to health workers to diagnose, treat, and counsel

**Table 2. Correlates of postpartum contraceptive use.**

| Factor | Unadjusted OR (95% CI) | Adjusted OR (95%CI) |
|---|---|---|
| Intercept | | 2.90 (0.22, 82.3) |
| Age of mother (ref = 15–24 years) | | |
| 25–34 years | 1.39 (0.95, 2.03) | 1.46 (0.95, 2.26) |
| 35–49 years | 0.90 (0.55, 1.46) | 1.00 (0.56, 1.78) |
| Pregnancy months at first ANC (ref = 1–3) | | |
| 4–9 months | 0.76 (0.50, 1.14) | 0.98 (0.62, 1.56) |
| Don't Know | 0.84 (0.31, 2.52) | 1.05 (0.35, 3.47) |
| Number of ANC visits (ref = 4 or more) | | |
| Less than 4 | 0.60 (0.42, 0.86)* | 0.72 (0.48, 1.08) |
| Don't Know | 0.79 (0.34, 2.01) | 0.79 (0.30, 2.17) |
| Place of delivery (ref = home/other) | | |
| Private | 1.44 (0.90, 2.31) | 0.20 (0.01, 2.21) |
| Public | 1.32 (0.88, 1.97) | 0.19 (0.01, 2.06) |
| Assistance with delivery (ref = skilled) | | |
| Unskilled | 0.70 (0.48, 1.02) | 0.16 (0.01, 1.74) |
| Number of FP methods known (ref = 0–3) | | |
| 4–6 | 1.61 (0.69, 3.84) | 1.89 (0.76, 4.77) |
| 7–9 | 2.83 (1.27, 6.40)* | 2.78 (1.19, 6.64)* |
| 10–12 | 3.53 (1.56, 8.14)* | 3.12 (1.31, 7.59)* |
| Discussed FP in last 12 months (ref = No) | | |
| Yes | 2.69 (1.89, 3.88)* | 2.58 (1.73, 3.86)* |
| FP information at ANC (ref = No) | | |
| Yes | 0.85 (0.60, 1.18) | 0.55 (0.36, 0.83) |
| FP service during delivery (ref = No) | | |
| Yes | 1.38 (0.98, 1.94) | 1.07 (0.69, 1.65) |
| FP service during PNC (ref = No) | | |
| Yes | 2.15 (1.52, 3.04)* | 2.04 (1.30, 3.24)* |
| FP service during immunization visits (ref = No) | | |
| Yes | 1.56 (1.11, 2.20)* | 0.96 (0.58, 1.60) |
| FP service during Well Child visits (ref = No) | | |
| Yes | 1.51 (1.07, 2.13)* | 0.91 (0.54, 1.53) |
| Pregnancy Intent (ref = No) | | |
| Undecided | 0.75 (0.49, 1.16) | 0.69 (0.42, 1.11) |
| Yes | 1.25 (0.85, 1.84) | 1.30 (0.83, 2.06) |

Note

* = statistically significant result; **FP** = Family Planning; **ANC** = Antenatal Care; **PNC** = Postnatal Care

women on all aspects of their pregnancies (e.g., test for underlying conditions, treat infections such as malaria, provide supplements such as iron folate and advice mothers on nutritional requirements) [29].

Various health facility points including ANC, PNC, immunization or child health services offer valuable, reliable opportunities along the continuum of care for healthcare providers to reach women at risk of closely spaced pregnancies with FP counseling and services [4]. They provide an opportunity to assess a woman's fertility needs and offer solutions to prevent unplanned pregnancies. In this study, women who received FP information at the immunization service delivery were 1.56 times more likely to use contraceptives than those who did not

receive these services. During the postpartum period, there are multiple contacts between women and healthcare providers when women are seeking child immunization services, yet the unmet need for contraception is still high [23]. These findings suggest that service delivery points for child welfare clinics including immunizations enable health workers to share information on contraceptive use, and consequently contributed to contraceptive uptake. In this study, we found that women who received FP information/ services during well-child visits were 1.5 times more likely to use contraception when compared to those who did not receive FP information/services during well-child visits. Jalang'o and colleagues established that more than three quarter (86.3%) of the respondents were found to have adopted postpartum contraception during the first year after delivery, 34.1% of the women with no intent to have more children were not using any contraceptive methods [18]. Use of contraceptives was driven by information from healthcare workers.

The finding that receipt of FP information during delivery is not associated with contraceptives contradicts findings by Jalang'o and colleagues [18] in which uptake of postpartum FP was significantly associated with counseling of women on various types of contraceptives. Findings from a similar study in Uganda showed that there are still missed opportunities for FP service provision at the mother-baby care points despite integration of services at such outlets [30]. As such even when postpartum women have numerous encounters with healthcare providers, they are unable to receive timely FP services. If available, the variable "time taken" between point of delivery and use of FP would have helped to determine factors that delay use of FP. A study in Uganda found that women who delivered at a government health facility had a shorter time to adoption of modern contraception compared with those who had births at home [31]. In addition, some women may never return to the facility or may return when they have conceived, contributing to the low overall CPR for the country.

We found that knowledge on different contraceptive methods was associated with postpartum contraceptive use. For example, women who knew more contraceptive methods were likely to use contraceptives. This finding coincides with findings from prior studies indicating that poor accessibility of contraceptive information or lack of awareness of contraceptive methods were identified as barriers to postpartum contraceptive use [17, 32]. Additionally, knowledge of the various contraceptive methods is influenced by factors such as media exposure [33], place of residence, modern contraceptive ever used, place of delivery, and FP counseling during PNC [34].

We acknowledge a number of limitations about this study. First, our results may not be generalizable to other settings as we conducted the study among women attending maternal and child health clinics in one county in Kenya. Because we selected participants in a non-random manner, we may have missed some women who did not come to the clinic for services such as immunizations, or those who receive their health care in other settings. Second, the cross-sectional nature of our data set prevents making causal inferences for our findings. Third, we did not consider a number of community-level variables, which in turn, may call for exploring multilevel modeling of postpartum contraceptive use. Fourth, the women surveyed in this study provided self-reported responses. Such data may be subject to recall bias and/or social desirability bias. Despite these limitations, these analyses provide valuable insights on potential determinants of contraceptive use in Kitui County, Kenya.

In conclusion, this study updates the current knowledge on determinants of contraceptive use in Kitui County in Kenya. There is an urgent need to bridge the gap between policy and implementation by increasing contraceptive use in Kitui County, currently standing at 68%. It is evident from the study that facility-level factors such as discussing FP with women; educating women about different FP methods; providing FP information or services during PNC, immunization, or well child visits are critical in increasing women's uptake of contraceptives

during postpartum period. Programs targeting these facility-level factors, for instance, those focused on enhancing women's attendance at PNC clinics, immunization, or well child visits should be encouraged.

## Acknowledgments

The authors would like to acknowledge Alexandra Mayer-Hohdahl—USAID KEA for reviewing this manuscript.

The views and opinions expressed in this paper are those of the author and not necessarily the views and opinions of the US Agency for International Development (USAID).

## Author Contributions

**Conceptualization:** Lilian Mutea, Immaculate Kathure, Damazo T. Kadengye, Sila Kimanzi, Daniel Wacira, Hesborn Wao.

**Data curation:** Immaculate Kathure, Damazo T. Kadengye, Nelson Onyango.

**Formal analysis:** Lilian Mutea, Immaculate Kathure, Damazo T. Kadengye, Daniel Wacira, Nelson Onyango, Hesborn Wao.

**Funding acquisition:** Nelson Onyango.

**Investigation:** Lilian Mutea.

**Methodology:** Lilian Mutea, Immaculate Kathure, Damazo T. Kadengye, Daniel Wacira, Hesborn Wao.

**Project administration:** Lilian Mutea.

**Software:** Nelson Onyango.

**Supervision:** Lilian Mutea, Hesborn Wao.

**Validation:** Lilian Mutea, Nelson Onyango.

**Visualization:** Lilian Mutea.

**Writing – original draft:** Lilian Mutea, Immaculate Kathure, Damazo T. Kadengye, Sila Kimanzi, Daniel Wacira, Hesborn Wao.

**Writing – review & editing:** Lilian Mutea, Immaculate Kathure, Damazo T. Kadengye, Sila Kimanzi, Daniel Wacira, Nelson Onyango, Hesborn Wao.

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
