## [Decision Letter · Decision Letter 0]

21 Dec 2021

PGPH-D-21-00638

Determinants of Contraceptive Use Among Women 0-23 Months Postpartum in Kitui County, Kenya: A Cross-Sectional Study

Dear Dr. Mutea,

Thank you for submitting your manuscript to PLOS Global Public Health. After careful consideration, we feel that it has merit but does not fully meet PLOS Global Public Health’s publication criteria as it currently stands. Therefore, we invite you to submit a revised version of the manuscript that addresses the points raised during the review process.

We have now received the reviews for your paper. Please revise your paper based on the comments of the reviewers. 

Please submit your revised manuscript 10th January 2022. If you will need more time than this to complete your revisions, please reply to this message or contact the journal office at globalpubhealth@plos.org. Please include the following items when submitting your revised manuscript:

We look forward to receiving your revised manuscript.

Kind regards,

Ajay Bailey, Ph. D.

Academic Editor

Journal Requirements:

1. Please include additional information regarding the survey or questionnaire used in the study and ensure that you have provided sufficient details that others could replicate the analyses. For instance, if you developed a questionnaire as part of this study and it is not under a copyright more restrictive than CC-BY, please include a copy, in both the original language and English, as Supporting Information.

2. Please amend your detailed Financial Disclosure statement. This is published with the article, therefore should be completed in full sentences and contain the exact wording you wish to be published.

i). Please include all sources of funding (financial or material support) for your study. List the grants (with grant number) or organizations (with url) that supported your study, including funding received from your institution. 

ii). State the initials, alongside each funding source, of each author to receive each grant.

iii). State what role the funders took in the study. If the funders had no role in your study, please state: “The funders had no role in study design, data collection and analysis, decision to publish, or preparation of the manuscript.”

iv). If any authors received a salary from any of your funders, please state which authors and which funders.

3. Please ensure that the funders and grant numbers match between the Financial Disclosure field and the Funding Information tab in your submission form. Note that the funders must be provided in the same order in both places as well.

4. Please update the completed 'Competing Interests' statement. If you have no competing interests to declare, please state “The authors have declared that no competing interests exist”.

5. Please ensure that you refer to Table 2 in your text as, if accepted, production will need this reference to link the reader to the Table.

Additional Editor Comments (if provided):

Dear Authors,

Thanks for your patience. We now have two reviews for your manuscript. Could you please makes the suggested changes and resubmit the paper?

Please indicated how you have taken the comments of the reviewers into account.

Best,

Ajay

Reviewers' comments:

Reviewer's Responses to Questions

**Comments to the Author**

1. Does this manuscript meet PLOS Global Public Health’s publication criteria? Is the manuscript technically sound, and do the data support the conclusions? The manuscript must describe methodologically and ethically rigorous research with conclusions that are appropriately drawn based on the data presented.

Reviewer #1: Yes

Reviewer #2: Yes

2. Has the statistical analysis been performed appropriately and rigorously?

Reviewer #1: Yes

Reviewer #2: Yes

3. Have the authors made all data underlying the findings in their manuscript fully available (please refer to the Data Availability Statement at the start of the manuscript PDF file)?

Reviewer #1: Yes

Reviewer #2: Yes

4. Is the manuscript presented in an intelligible fashion and written in standard English?

Reviewer #1: Yes

Reviewer #2: Yes

5. Review Comments to the Author

Reviewer #1: I am pleased going through the article. The topic is critical dealing with barriers in using early use of contraceptive methods in case of Kenya. However, I a few following suggestions to enhance the utility of the work.

1. Since the contraceptive prevalence rate is at 68%, how much further increase authors expect by the policy recommendations mentioned in this research. I would suggest that authors should highlight and find the early use of FP method after the child birth. The early use is critical otherwise there are higher chance of unintended pregnancy or even intended pregnancy resulting in birth with shorter intervals. Such consequences have detrimental effects on maternal and child health. Therefore, I suggest to use the information on the gap between the childbirth and the first of family planning method in the logistic regression analysis.

2.In the the method section, authors should illustrate (from MCH & FP services guidelines of the government documents) that what kind of issues get covered in the FP discussion with women.

3. Table 1, Authors should show chi-square with p-values so that these frequencies distributions would not be misled as random.

Editorial suggestion:

4. Abstract says women having child within 0-59 months where following write-up says 0-23 months. It is creating confusion, so make it coherent throughout the article.

5. Similarly, Well Child visit and Well-baby clinic visit are used interchangeably, again make the language consistent.

Reviewer #2: Article: Manuscript Number: PGPH-D-21-00638

Full Title: Determinants of Contraceptive Use Among Women 0-23 Months Postpartum in Kitui County, Kenya: A Cross-Sectional Study Reviewer’s feedback

This is an important topic to study to examine determinants of Contraceptive Use Among Women 0-23 Months Postpartum in Kitui County, Kenya. This may add value to the literature in terms of study variables. However, the following are the feedback that can be considered.

Introduction section:

Although you have cited the WHO recommendation for birth intervals of between 2 and 3 years. On other hand it was mentioned with literature support that “the majority (91%) of postpartum women in low- and middle-income countries (LMIC) desire to space birth by at least a year [11], contraceptive uptake continues to be low in this region [12, 13].

Suggestion: it will be good to add critical discussion here the reason for the desire to space birth by at least a year.

Methodology: under the research methods, it will be important to mention the method of selecting study participants. How the setting has been selected and why?

The study findings are expected to inform the design of effective

On page 4, it was mentioned that the “strategies aimed at increasing contraceptive use and further increase the CPR and make recommendations on effective strategies for increasing contraceptive use among postpartum”. Suggestion: Make sure the study aim marries well with the purpose of your study.

Results: in the result section it was mentioned that 41% of participants (non-users) do not discuss FP. Suggestion: This is an interesting finding and can further be discussed critically in the discussion chapter.

Discussion:

It was discussed that “the likelihood of contraceptive use increased with the increase in the number of FP methods known. For example, women who knew 7-9 FP methods were 2.78 times more likely to use contraceptives than those who knew 0 to 3 FP methods.

Suggestion: it will add more value to this paper if authors may specify the variables that are relevant to the knowledge regarding FP and then bring in the local knowledge about why some of the participants still don’t acquire FP knowledge?

Regarding the discussion on “women who received FP information/ services during well-child visits had elevated odds of contraceptive use (OR = 1.51, 95%CI: 1.07, 2.13)”.

Suggestion: please make it more explicit and unfold about elevated odds of contraceptive use.

Hope that helps!

Overall great efforts!

All the best!

6. PLOS authors have the option to publish the peer review history of their article (what does this mean?). If published, this will include your full peer review and any attached files.

**Do you want your identity to be public for this peer review?** For information about this choice, including consent withdrawal, please see our Privacy Policy.

Reviewer #1: **Yes: **Chander Shekhar, Professor, Department of Fertility & Social Demography, International Institute for Population Sciences, Mumbai, India

Reviewer #2: **Yes: **Saleema Gulzar

---

## [Editor Report · Decision Letter 1]

21 Apr 2022

Determinants of Contraceptive Use Among Women 0-23 Months Postpartum in Kitui County, Kenya: A Cross-Sectional Study

PGPH-D-21-00638R1

Dear Ms Mutea,

We are pleased to inform you that your manuscript 'Determinants of Contraceptive Use Among Women 0-23 Months Postpartum in Kitui County, Kenya: A Cross-Sectional Study' has been provisionally accepted for publication in PLOS Global Public Health.

Best regards,

Ajay Bailey, Ph. D.

Academic Editor